# Beneficial Effects of a Low-Nickel Diet on Relapsing IBS-Like and Extraintestinal Symptoms of Celiac Patients during a Proper Gluten-Free Diet: Nickel Allergic Contact Mucositis in Suspected Non-Responsive Celiac Disease

**DOI:** 10.3390/nu12082277

**Published:** 2020-07-29

**Authors:** Raffaele Borghini, Natascia De Amicis, Antonino Bella, Nicoletta Greco, Giuseppe Donato, Antonio Picarelli

**Affiliations:** 1Department of Translational and Precision Medicine, Sapienza University, 00185 Rome, Italy; raffaele.borghini@uniroma1.it (R.B.); n.deamicis@libero.it (N.D.A.); nicoletta.greco25@gmail.com (N.G.); giuseppe.donato@uniroma1.it (G.D.); 2Department of Infectious Diseases, Istituto Superiore di Sanità, 00161 Rome, Italy; antonio.bella@iss.it

**Keywords:** celiac disease, refractory celiac disease, remission, gluten-free diet, nickel allergy, allergic contact mucositis, irritable bowel syndrome (IBS), low-nickel diet

## Abstract

Background and Aim: Nickel (Ni)-rich foods can induce allergic contact mucositis (ACM) with irritable bowel syndrome (IBS)-like symptoms in predisposed subjects. Ni ACM has a high prevalence (>30%) in the general population and can be diagnosed by a Ni oral mucosa patch test (omPT). Many celiac disease (CD) patients on a gluten-free diet (GFD) often show a recrudescence of gastrointestinal and extraintestinal symptoms, although serological and histological remission has been achieved. Since a GFD often results in higher loads of ingested alimentary Ni (e.g., corn), we hypothesized that it would lead to a consequent intestinal sensitization to Ni in predisposed subjects. We wanted to (1) study Ni ACM prevalence in still symptomatic CD patients on a GFD and (2) study the effects of a low-Ni diet (LNiD) on their recurrent symptoms. Material and Methods: We recruited 102 consecutive CD patients (74 female, 28 male; age range 18–65 years, mean age 42.3 ± 7.4) on a GFD since at least 12 months, in current serological and histological remission (Marsh–Oberhuber type 0–I) who complained of relapsing gastrointestinal and/or extraintestinal symptoms. Inclusion criteria: presence of at least three gastrointestinal symptoms with a score ≥5 on the modified Gastrointestinal Symptom Rating Scale (GSRS) questionnaire. Exclusion criteria: IgE-mediated food allergy; history of past or current cancer; inflammatory bowel diseases; infectious diseases including *Helicobacter pylori*; lactose intolerance. All patients enrolled underwent Ni omPT and followed a LNiD for 3 months. A 24 symptoms questionnaire (GSRS modified according to the Salerno Experts’ Criteria, with 15 gastrointestinal and 9 extraintestinal symptoms) was administered at T0 (free diet), T1 (GFD, CD remission), T2 (recurrence of symptoms despite GFD), and T3 (GFD + LNiD) for comparisons. Comparisons were performed using Wilcoxon signed-rank test. RESULTS: Twenty patients (all female, age range 23–65 years, mean age 39.1 ± 2.9) out of 102 (19.6%) were finally included. All 20 patients enrolled (100%) showed positive Ni omPT, confirming an Ni ACM diagnosis. A correct GFD (T0 vs. T1) induced the improvement of 19 out of the total 24 (79.2%) symptoms, and 14 out of 24 (58.3%) were statistically significant (*p*-value < 0.0083 according to Bonferroni correction). Prolonged GFD (T1 vs. T2) revealed the worsening of 20 out of the total 24 (83.3%) symptoms, and 10 out of 24 (41.7%) were statistically significant. LNiD (T2 vs. T3) determined an improvement of 20 out of the total 24 (83.4%) symptoms, and in 10 out of 24 (41.7%) symptoms the improvement was statistically significant. Conclusions: Our data suggest that the recrudescence of gastrointestinal and extraintestinal symptoms observed in CD subjects during GFD may be due to the increase in alimentary Ni intake, once gluten contamination and persisting villous atrophy are excluded. Ni overload can induce Ni ACM, which can be diagnosed by a specific Ni omPT. Improvement of symptoms occurs after a proper LNiD. These encouraging data should be confirmed with larger studies.

## 1. Introduction

Celiac disease (CD) is a chronic inflammatory bowel disease triggered by the ingestion of gluten in genetically susceptible individuals, who test positive for human leukocyte antigen (HLA) DQ2 and/or DQ8. Its prevalence is about 1%, and since the small intestine is its main target organ, CD can have gluten-related gastrointestinal manifestations, such as bloating, abdominal pain, diarrhea, and constipation [1,2,3]. What is more, CD is a multisystem disorder, and patients can also complain of extraintestinal signs and symptoms [4,5]. CD diagnosis in adults is usually based on positive results of specific serological tests for anti-endomysial antibodies (EMA) and anti-tissue transglutaminase (tTG) antibodies performed during a free diet and then confirmed by the finding of intestinal villous atrophy on histological examination of duodenal biopsies. The only treatment currently available is a lifelong and strict gluten-free diet (GFD) but, nevertheless, many CD patients complain about the persistence or relapse of symptoms even during GFD [6]: in this case, interviews with gastroenterologists and nutritionists are necessary in order to investigate a proper adherence to the GFD; moreover, repetition of serological tests and duodenal biopsies are mandatory to exclude ongoing intestinal damage and gluten exposure. When persistent damage in the duodenal mucosa is found despite a correct GFD, refractory CD and possible complications such as intestinal lymphoma must be investigated [7].

Moreover, in clinically non-responding CD, other possible overlapping diagnoses should be considered, such as inflammatory bowel diseases (IBD), and irritable bowel syndrome (IBS)-like disorders (e.g., lactose intolerance), but many cases seem to remain unsolved [8]. Recently, a diet low in fermentable oligo-, di-, and monosaccharides and polyols (FODMAPs) has been proposed as an ex adiuvantibus treatment to reduce IBS-like symptoms in CD patients following a GFD, although there is no specific indication or supporting diagnostic test [9,10].

More recently, nickel (Ni) allergic contact mucositis (ACM), which is linked to the ingestion of Ni-rich foods, has been added to IBS-like disorders. Together with Ni allergic contact dermatitis (ACD), Ni ACM is an expression of “systemic Ni allergy syndrome” (SNAS) and can have both gastrointestinal and extraintestinal manifestations. According to the European Surveillance System on Contact Allergy (ESSCA), the prevalence of an epicutaneous patch test positive to Ni may reach 30% in some European countries, but Ni ACM prevalence may even be higher [11]. Patients affected by Ni ACM show a low-grade intestinal inflammation with a local adaptive response to Ni-containing foods: this mucositis seems to be characterized by increased lymphocyte trafficking (type IV immune response) [12,13]. Ni ACM diagnosis is currently based on a Ni oral mucosa patch test (omPT), which has already proved good sensitivity and specificity [13], and a low-Ni diet (LNiD) can be thus suggested in this condition in order to significantly reduce both Ni-related gastrointestinal and extraintestinal symptoms [14,15,16,17,18,19].

Figure 1 shows the main foods with the highest Ni content, and it is easy to observe that many of them (e.g., corn) are consumed in large quantities by CD patients on a proper and strict GFD [15,16]. It is therefore possible that a high load of alimentary Ni may induce or exacerbate a “Ni sensitivity” in predisposed subjects, especially in CD patients on a long-term GFD.

On these premises, our aims were (1) to study the prevalence of Ni ACM in CD patients in serological and histological remission with relapsing symptoms; (2) to evaluate the effects of an LNiD on gastrointestinal and extraintestinal symptoms in these patients.

## 2. Materials and Methods

### 2.1. Patients

Study design: pilot study. We recruited 102 consecutive CD patients (74 female, 28 male; age range 18–65 years, mean age 42.3 ± 7.4) on a GFD since at least 12 months, with current serum EMA and anti-tTG antibodies negative results and histological remission (Marsh–Oberhuber type 0–I) who complained of relapsing or persisting gastrointestinal and/or extraintestinal symptoms. They referred to our Gastroenterology Unit from January 2017 to December 2019. Their CD diagnosis had been previously made according gluten-related signs and symptoms, serum EMA and anti-tTG antibodies positive results, and the histological finding of duodenal villous atrophy (Marsh–Oberhuber type IIIA, B, or C) [20,21,22].

Inclusion criteria: presence of at least three gastrointestinal symptoms with a score ≥5 on the modified GSRS questionnaire completed at study recruitment, in order to exclude patients with less significant clinical pictures. Exclusion criteria: IgE-mediated food allergy (diagnosed by skin prick test or laboratory tests (ImmunoCAP) for serum allergen-specific IgE antibodies); history of past or current cancer; inflammatory bowel disease; infectious diseases including *Helicobacter pylori* (HP); lactose intolerance (by means of the lactose breath test and genetic evaluation of lactase-gene polymorphism [23,24]).

The study was performed in compliance with the Declaration of Helsinki. Approval of the local ethics committee was obtained (study approval: report 8.2.0 06/2020 of the Board of the Department of Translational and Precision Medicine—Sapienza University of Rome). Written informed consent was obtained from all patients.

### 2.2. Symptom Questionnaire

The Gastrointestinal Symptom Rating Scale (GSRS) questionnaire modified according to “Salerno’s experts’ criteria” is a standardized tool used in the diagnostic protocol for non-celiac gluten sensitivity, and it has also been employed to objectively evaluate the clinical status in other IBS-like disorders such as Ni sensitivity [15]. It consists of a list of gastrointestinal and extraintestinal symptoms associated to a numeric scale (score ranging from 0 to 10), which represents the intensity perceived by the patients during a specific dietary regimen [25]. Gastrointestinal symptoms include abdominal pain, heartburn, acid regurgitation, bloating, nausea, borborygmus, swelling, belching, flatulence, decreased or increased evacuations, loose or hard stools, urgent need for defecation, oral/tongue ulcers. Extraintestinal symptoms include dermatitis, headache, foggy mind, fatigue, numbness of the limbs, joint/muscle pain, fainting [15].

According to our standard outpatient treatment protocol for the management of CD, the questionnaire had been previously administered at CD diagnosis (T0) and after at least 12 months of GFD, when serological and histological remission had been achieved (T1). After at least three further months on a GFD, a third questionnaire was administered to those CD patients who complained of a relapse of gastrointestinal and extraintestinal symptoms, despite confirmed negative serological and histological results to exclude refractory CD (T2, study recruitment). The last questionnaire was administered after 3 months of LNiD in addition to GFD (T3).

### 2.3. Nickel Oral Mucosa Patch Test

Once enrolled in the study (T2), all patients underwent an Ni omPT to detect the presence of Ni ACM. Ni omPT is a 5 mm filter paper disk saturated with a 5% solution of Ni sulfate in Vaseline (0.4 mg Ni-sulfate/8 mg Vaseline). It is applied on the upper lip mucosa and held in place by a transparent adhesive film (Tegaderm, 3M, St. Paul, MN - USA). For appropriate diagnostic purposes, a control test with only 8 mg Vaseline is also provided and applied. Local Ni-induced type IV hypersensitivity reactions (e.g., edema, hyperemia, aphthous/vesicular lesions) can be evident after just 2 h of exposure or even after 24–48 h as late reactions. Late general symptoms triggered by omPT (e.g., swelling, abdominal pain, diarrhea, headache, foggy mind, itching) should also be considered as positive test results [14,15].

### 2.4. Low-Nickel Diet

All enrolled patients followed a balanced GFD with the addition of an LNiD for 3 months after a visit with trained dieticians, who verified correct adherence to both diet regimens by means of a daily dietary diary and biweekly telephone interviews. Ni is an element abundantly present in many foods, with a certain concentration variability depending on the type of soil and plant species, irrigation water, fertilizers, and pesticides. Thus, since its total elimination from the diet is impossible, we recommended to avoid only foods with an estimated high content of Ni (Ni > 100 μg/kg) (Figure 1). The use of stainless-steel utensils and pots has also been discouraged, in order to reduce Ni contamination during cooking [15,19,26].

### 2.5. Statistical Analysis

Data obtained during the present study were both qualitative (omPT results) and quantitative (modified GSRS questionnaire). Qualitative data were expressed as frequencies (both absolute and relative). The symptoms’ scores (GSRS scale: 0 = absent, 10 = maximum intensity) were summarized by median, and Wilcoxon’s signed-rank test was used to compare each symptom at different times (T0, T1, T2, T3). Applying the Bonferroni correction, *p*-value <0.0083 (alpha = 0.05/6 comparisons) was considered statistically significant. Statistical analysis was performed using the Stata software, version 16.0 (Stata Cooperation, College Station, TX, USA).

Study arrangement and patient enrollment are summarized in Figure 2a,b.

Study management, Ni omPT, administration of the modified GSRS questionnaire, patient follow-up, and final data processing were performed at the Department of Translational and Precision Medicine.

## 3. Results

### 3.1. Patients

Of the 102 patients recruited, 17 patients were excluded since they did not meet the criterion of at least three gastrointestinal symptoms with a score ≥5 in the GSRS questionnaire completed at T2. Sixty-two out of the remaining 85 patients were also excluded: 54 were lactose intolerant, 7 were affected by HP infection, 1 was affected by overlapping active ulcerative colitis. Three out of the remaining 23 patients dropped out of the study, reporting that they no longer wanted to follow further food restrictions. Therefore, a total of 20 patients (all female, age range 23–65 years, mean age 39.1 ± 2.9, median age 40) completed the study (Figure 2b).

### 3.2. Nickel Oral Mucosa Patch Test

All 20 patients studied (100%) showed Ni omPT positive results and received an Ni ACM diagnosis. They all showed evident local mucosal alterations induced by Ni (erythema, edema, and/or vesicles) within 2 h after patch application (Figure 3a,b). What is more, all 20 patients showed at least one additional gastrointestinal or extraintestinal systemic symptom within 48 h after Ni omPT.

### 3.3. Symptom Questionnaire

A correct GFD (T0 vs. T1) induced an improving trend in 19 out of the total 24 (79.2%) symptoms, and 14 out of 24 (58.3%) were statistically significant (*p*-value < 0.0083).

The prolonged GFD (T1 vs. T2) revealed a worsening trend in 20 out of the total 24 (83.3%) symptoms, and 10 out of 24 (41.7%) were statistically significant: abdominal pain, bloating, nausea, swelling, loose stools, dermatitis, fatigue, numbness of the limbs, and muscle and joint pain.

Once an Ni ACM diagnosis was obtained, an LNiD (T2 vs. T3) determined an improving trend in 20 out of the total 24 (83.4%) symptoms, and in 10 out of 24 (41.7%) symptoms the improvement was statistically significant. In detail, 12 out of 15 (80%) gastrointestinal symptoms improved, and 7 out of 15 (46.7%) showed a statistically significant improvement. In the same interval, 8 out of 9 (88.9%) extraintestinal symptoms showed an improvement, and 3 out of 9 (33.3%) significantly improved.

More details about gastrointestinal and extraintestinal symptoms during the different intervals analyzed are reported in Figure 4 and Figure 5 and Table 1.

## 4. Discussion

The persistence or recurrence of gastrointestinal and/or extraintestinal symptoms in CD patients during GFD is a very common condition and is a topic of great relevance. The causes of this problem are to be initially searched in an incorrect adherence to GFD and this is what can happen in those patients who still show persistently positive antibody titers and significant duodenal histological alterations, even despite quite a long period of GFD (>12 months). Furthermore, the possibility of refractory CD is always to be taken into consideration [4,6,7,8].

This issue becomes even more difficult to decipher and solve when CD patients reach serological and histological remission, but symptoms are still present or show a new peak despite a correct GFD. Furthermore, some of these patients may even report the appearance of new symptoms never complained about before. Other overlapping disorders, such as IBS or IBS-like disorders may be the causes of these symptoms and, in this regard, encouraging results have been obtained by ex adiuvantibus use of a low-FODMAP diet, although this approach revealed some limitations that will be discussed further [9,10].

On the other hand, many gluten-free foods consumed by CD patients are high in Ni content. Therefore, once a CD diagnosis has been obtained, the progressive Ni load induced by the GFD can trigger a relapse of symptoms in subjects predisposed to Ni allergy. Ni is present in many foods with different concentrations. It can be responsible for SNAS, which can have both gastrointestinal and extraintestinal manifestations. Specifically, Ni ACM is estimated to be one of the most common IBS-like disorders, and its diagnosis can rely on an Ni omPT, more sensitive and specific than the epicutaneous patch test [11]. What is more, excellent clinical results have already been obtained by Ni omPT and LNiD in the management of IBS-like and extraintestinal symptoms of women suffering from endometriosis who were still symptomatic despite different treatments [15,17].

Based on these considerations, we investigated the prevalence of Ni ACM in still symptomatic CD subjects after appropriate GFD and studied the effects of an LNiD on their gastrointestinal and extraintestinal symptoms.

Firstly, we selected symptomatic CD patients on a proper GFD who had no more serological and histological signs of disease activity. Then, we excluded those who did not meet the minimum clinical criterion by means of the GSRS questionnaire: in this way, we eliminated the less disabling and most confounding clinical pictures, even if this has led to a reduction in the number of patients studied. Other possible overlapping confounding pathologies have been excluded, such as lactose intolerance, HP infection and IBDs.

Our results showed an Ni ACM prevalence of 100% in the final 20 patients actually enrolled (Figure 3a,b): this percentage may appear extraordinarily high, but it is mandatory to consider not only the high prevalence of Ni ACM in the general population (estimated to be even greater than 30%) but also the strict exclusion criteria previously applied [11]. These 20 patients with Ni omPT positive results should be contextualized among the 85 CD patients who had a significant symptomatic picture: thus, Ni ACM should have a prevalence of at least 23.5% in our study. This percentage could have been higher considering not only the dropouts but also those patients affected by other pathologies who had been excluded in recruitment phase: in fact, Ni ACM can also easily overlap with other disorders, especially lactose intolerance, and in our study we excluded 54 lactose intolerant patients (about 63.5% of the 85 patients with a significant clinical picture) [11].

Afterward we focused on the effects of the different diet regimens on the symptoms.

First, we confirmed a general clinical improvement after CD diagnosis and a correct GFD (T0 vs. T1): once serological and histological remission were achieved, about 80% of the 24 total symptoms improved (almost 60% was also statistically significant).

On the other hand, a prolonged strict GFD (T1 vs. T2) resulted in a general clinical relapse involving more than 80% of all symptoms (the worsening was statistically significant in more than 40% of the symptoms). This negative change may be attributed to a GFD-related load of Ni in already sensitive or predisposed subjects.

This theory seemed to be confirmed after a balanced restriction of Ni-rich foods: in only three months, GFD plus LNiD (T2 vs. T3) induced an improvement of more than 80% of the symptoms and in the half of the cases the improvement was statistically significant, including the most complained about and disabling symptoms, such as abdominal pain, swelling, increased evacuations, and loose stools. Moreover, many of them got even better compared to the initial GFD alone (T1 vs. T3), although this difference was not statistically significant. As regards extraintestinal symptoms in the T2 vs. T3 range, dermatitis, headache, and fatigue statistically improved. Dermatitis deserves a special mention, as it showed a very peculiar trend: at the beginning (T0 and T1) it was almost totally absent, then it was significantly exacerbated reaching an acute peak at T2 and finally significantly reduced/resolved after GFD plus LNiD at T3. The curve of dermatitis’ clinical course can further suggest the interference of an “alimentary trigger factor” during prolonged GFD, which is not related to gluten contamination, as demonstrated by the negative serological and histological results. The dietary profile of CD patients on a GFD and the impressive results of both Ni omPT and LNiD, would confirm that alimentary Ni overload is able to unmask/exacerbate not only gastrointestinal but also systemic symptoms, over a medium to long-term time period.

It should be emphasized that no symptoms significantly improved or worsened by comparing T0 (free diet) with T2 (Ni overload during GFD), suggesting a close clinical similarity between these two times: this is what often makes gastroenterologists think that the cause of the clinical relapse is a new gluten contamination.

The comparison between free diet and GFD + LNiD (T0 vs. T3) led to an improving trend of more than 70% of the symptoms: this is a very good percentage, although slightly lower than the almost 80% obtained from GFD alone, before Ni overload (T0 vs. T1). This may mean that, although LNiD is very effective in achieving a new clinical remission, Ni-sensitive patients cannot completely eliminate Ni, therefore, Ni-related symptoms, from their GFD.

The comparison between the well-being obtained by the initial GFD alone and the well-being obtained by the addition of the LNiD (T1 vs. T3) is also interesting: these two stages showed no statistically significant difference in symptoms’ intensity perceived. This means that after the relapse peak in T2 (Ni load), a correct dietary intervention (LNiD) is able to completely restore well-being again.

The absence of a trial design was a limit of our study. Furthermore, it was carried out in a single center, the final sample size was quite small, and finally resulted in including only female patients. In addition, the very high prevalence of Ni omPT positive results (100%) may seem misleading.

Firstly, our results must be contextualized in the initial larger pool (102 total patients): the choice to exclude from the study those patients with less marked symptomatic pictures certainly led to the underdiagnosis of many other Ni-sensitive patients. In addition, it should be considered that Ni ACM can coexist with other disorders and can overlap with them from the symptomatic point of view [11]: in our study many HP-positive and lactose intolerant patients (almost 60% of the 102 patients initially recruited) were excluded for methodological correctness, and, thus, many other Ni-sensitive patients were probably lost among them.

The fact that the final 20 patients studied were only females is probably due to the greater prevalence of females in CD: in literature the female/male ratio is estimated to be about 3:1 and this proportion is approximately preserved in the 102 patients recruited at the beginning [27]. Furthermore, it has been described that Ni can act as a metalloestrogen and, thus, may have a greater influence in women with both extraintestinal and gastrointestinal clinical manifestations [15,28].

Given the strict differential diagnosis previously performed, this would explain such a high percentage (100%) of Ni omPT positive results. Moreover, the specific and successful treatment by LNiD seemed to confirm the appropriateness of our assumptions and supported the Ni omPT positive results: as above mentioned, more than 80% of symptoms improved after LNiD and about the half of them were statistically significant. In this regard, if we had not used the Bonferroni correction (*p* < 0.05 instead of *p* < 0.0083), the improvements of some other important symptoms (borborygmus, foggy mind, muscle pain, and joint pain) would have resulted statistically significant. We hope that future trials with larger populations will be able to confirm these preliminary observations.

Another weak point of our study, as well as possible obstacle for future trials, may be the impossibility to accurately measure Ni contained in foods and biological samples from patients studied. There is some variability of Ni content in foods and, to date, there are still no standard methods to measure it routinely: if these methods existed, we would have the possibility to prescribe highly personalized diets, more effectively monitor the intake of Ni-rich foods, and make even more appropriate comparisons. As recently demonstrated, we can successfully overcome this limit by prescribing patients a balanced LNiD on the base of an estimated average content of Ni in foods and under direct control of trained dieticians. A daily dietary diary and a detailed interview were also used to verify correct adherence to the GFD and LNiD.

It has already been described in literature that a low-FODMAP diet as an ex adiuvantibus treatment can benefit still symptomatic CD patients on a GFD. However, this dietary intervention has not so far been supported by specific diagnostic tests [9]. Moreover, it is known that many foods with estimated high content of FODMAPs may also cause other IBS-like disorders, such as lactose intolerance. It is therefore possible that during a low-FODMAP diet, other underlying and unrecognized diseases are treated. In this regard, we also observed a significant overlap between FODMAP-rich and Ni-rich foods (Figure 1), especially corn and other gluten-free foods consumed by CD subjects. It is therefore possible that the benefits of a low-FODMAP diet can depend on a concomitant involuntary LNiD in unrecognized Ni-sensitive patients. Given the high prevalence of Ni ACM and our results, this is more than only a hypothesis. On the other hand, our study can claim among its strengths an accurate preliminary differential diagnosis with other common IBS-like disorders and organic diseases. In addition, a targeted LNiD was prescribed after a reliable specific diagnostic test (Ni omPT), thus avoiding unnecessary dietary exclusions [15].

It may also be interesting to discuss the possible effects of close contact with patients during follow-up: frequent and extensive dietary interviews can have a placebo effect in clinical setting, capable even of inhibiting a symptom. On the other hand, they are essential methodological tools for an adequate quality assessment of the diet followed, as well as for the determination of the patient’s clinical status. Close clinical contact also seemed to play a relevant role in supporting such “delicate” patients who had to follow two strict diet regimens: GFD and LNiD. This was confirmed not only by the encouraging clinical results obtained but also by the very low number of drop-outs. In addition, close clinical contact appeared even more necessary for such a study spread over a long period of time: the long time span of observation could have led to misestimation of the beneficial or harmful effects of dietary interventions. Finally, the inclusion of a significant number of symptoms in the GSRS standardized test for clinical evaluation has most probably helped to further reduce possible placebo/nocebo effects and misestimation of the results.

## 5. Conclusions

In conclusion, our findings show for the first time that Ni-rich foods and Ni ACM can frequently be the cause of relapsing gastrointestinal and extraintestinal symptoms in CD patients, even/especially during a correct GFD. Furthermore, our study not only confirms the usefulness of Ni omPT in making an Ni ACM diagnosis but also highlights that a balanced LNiD in addition to a correct GFD can offer a significant clinical improvement in this category of patients.

Further studies with larger populations should be carried out to confirm these important data, which may change the clinical management of CD patients.

## Figures and Tables

**Figure 1 nutrients-12-02277-f001:**
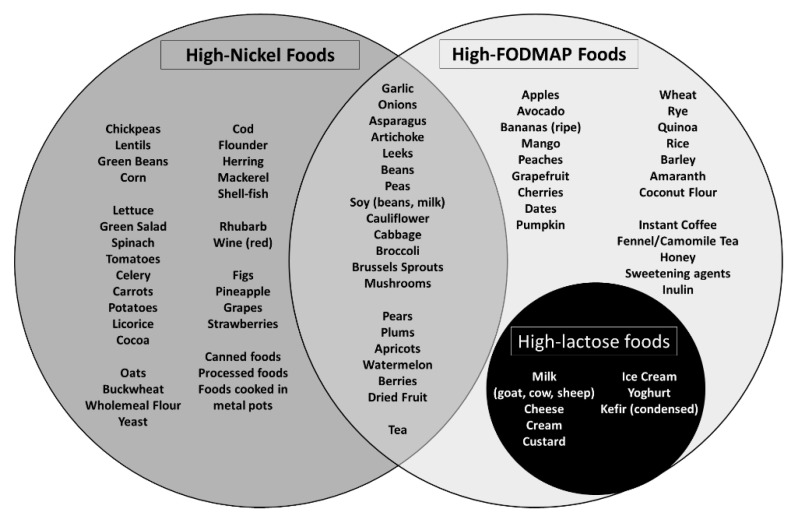
Foods with high nickel (Ni) content and their possible overlap with foods rich in fermentable oligo-, di-, and monosaccharides and polyols (FODMAPs). Here we report some of the main foods belonging to specific categories. To be noted is the overlap between Ni-rich foods and foods with high FODMAP content, as well as the overlap between foods high in FODMAPs and lactose content [10,15,16,17,18,19].

**Figure 2 nutrients-12-02277-f002:**
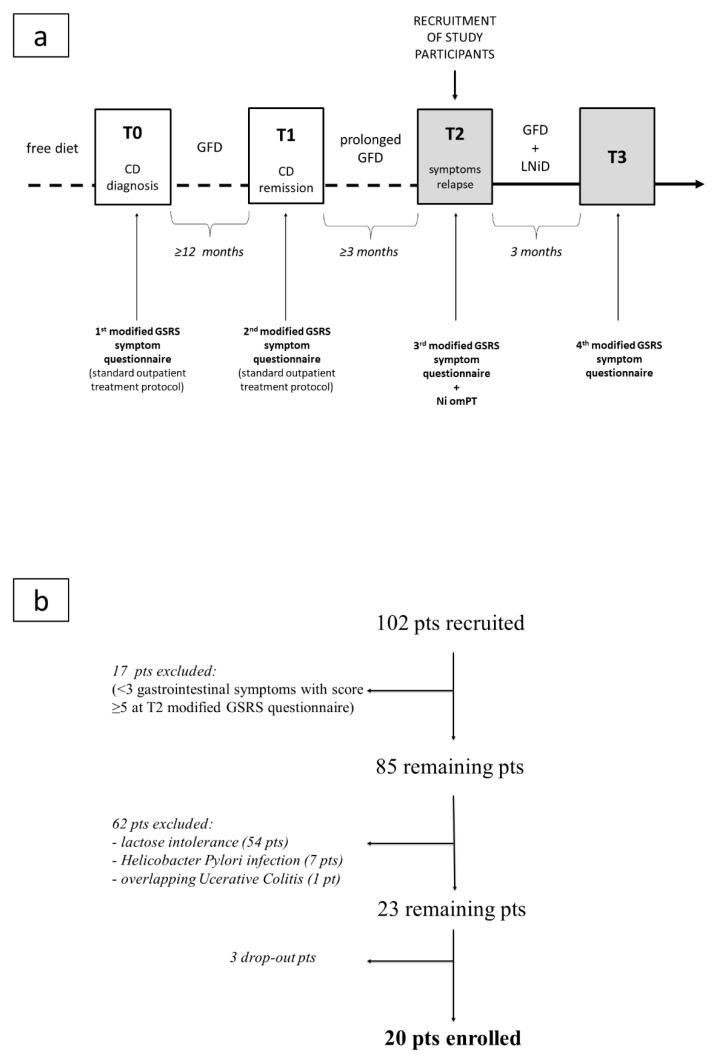
Flow charts of the study: (**a**) study arrangement; (**b**) patient enrollment. Legend: CD, celiac disease; GFD, gluten-free diet; GSRS, Gastrointestinal Symptom Rating Scale; LNiD, low-nickel diet; Ni omPT, nickel oral mucosa patch test; pts, patients.

**Figure 3 nutrients-12-02277-f003:**
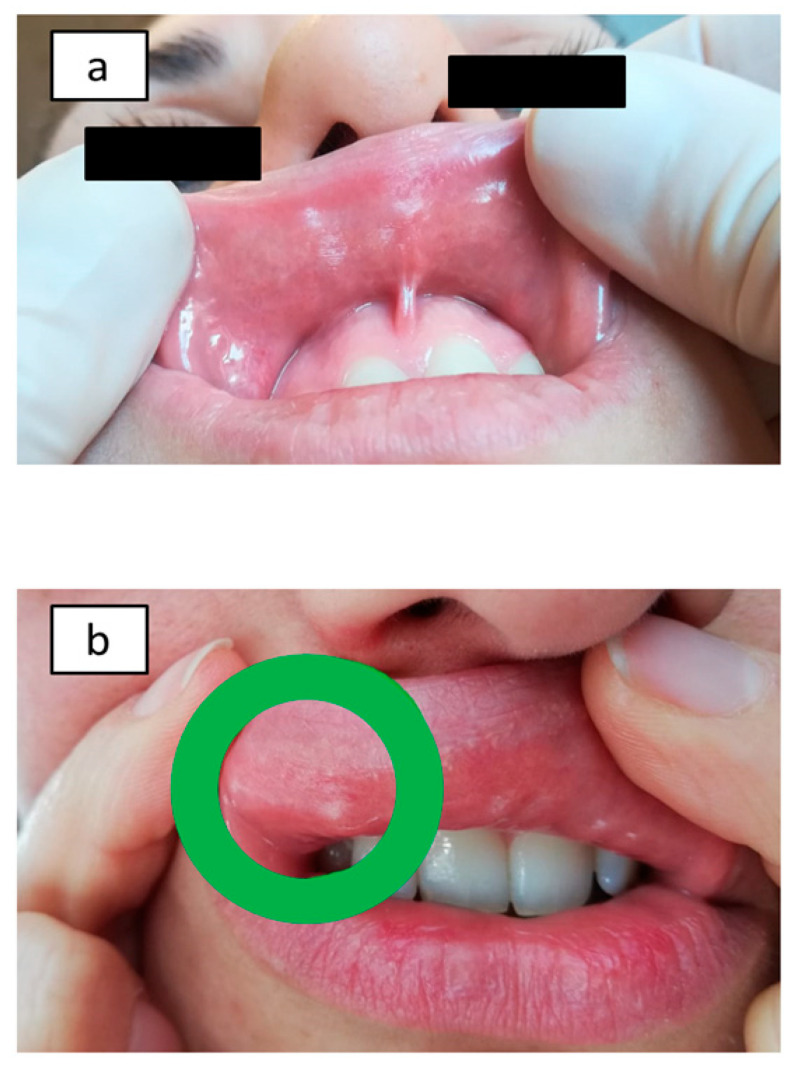
Nickel oral mucosa patch test (Ni omPT) results: Ni-sensitive patients before Ni omPT application (**a**) and after Ni omPT removal (2 h) (**b**).

**Figure 4 nutrients-12-02277-f004:**
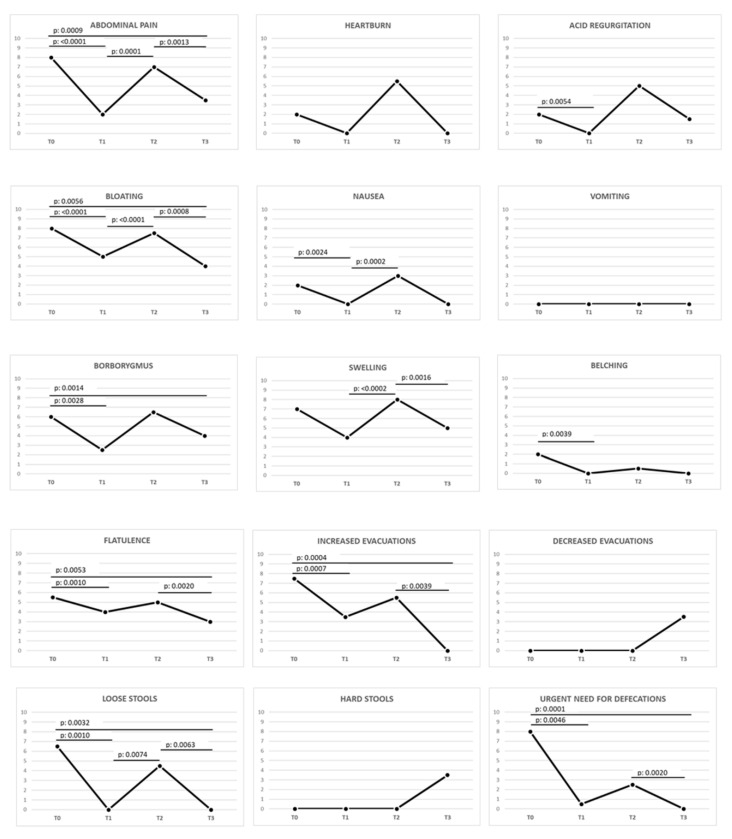
Variation of gastrointestinal symptoms in celiac patients during different stages of the study. The *p*-value was calculated using the Wilcoxon signed-rank test (statistically significant *p*-value < 0.0083 according to Bonferroni correction). Legend: GSRS, Gastrointestinal Symptom Rating Scale; T0, baseline, during gluten-containing diet; T1, after ≥12 months of proper gluten-free diet; T2, after ≥3 months of prolonged gluten-free and Ni-rich diet; T3, after 3 months of low-nickel and gluten-free diet.

**Figure 5 nutrients-12-02277-f005:**
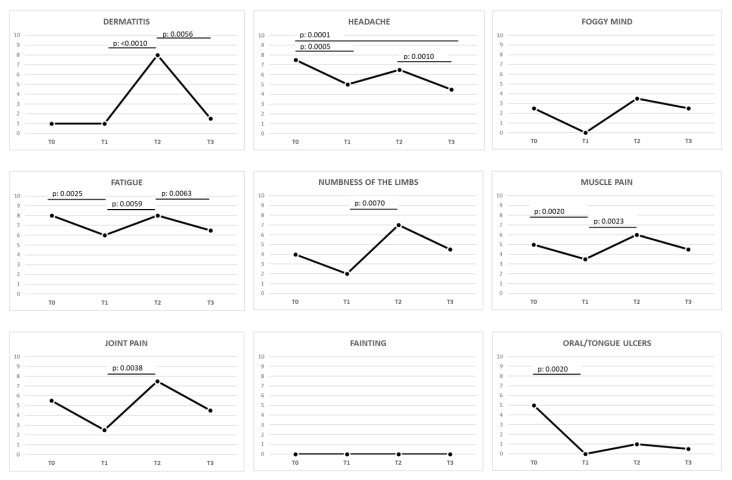
Variation of extraintestinal symptoms in celiac patients during different stages of the study. The *p*-value was calculated using the Wilcoxon signed-rank test (statistically significant *p*-value < 0.0083 according to Bonferroni correction). Legend: GSRS, Gastrointestinal Symptom Rating Scale; T0, baseline, during gluten-containing diet; T1, after ≥12 months of proper gluten-free diet; T2, after ≥3 months of prolonged gluten-free and nickel-rich diet; T3, after 3 months of low-nickel and gluten-free diet.

**Table 1 nutrients-12-02277-t001:** Gastrointestinal and extra-intestinal symptoms during the different intervals analyzed. The table shows for each category of symptoms how many of them improved, worsened or remained stable in the different ranges of time considered. Data are reported in both absolute and percentage values. Legend: CD, celiac disease; GFD, gluten-free diet; LNiD, low nickel diet; n, number; Ni, Nickel; STAT. SIGN., statistically significant; tot, total; VS, versus.

	Comparison Description	Gastrointestinal Symptoms (tot *n* = 15)					Extra-Intestinal Symptoms (tot *n* = 9)					All Symptoms (tot *n* = 24)				
		Improvement		Worsening		Stability *n* (%)	Improvement		Worsening		Stability *n* (%)	Improvement		Worsening		Stability *n* (%)
		*n* (%)	Stat. Sign. *n* (%)	*n* (%)	Stat. Sign. *n* (%)		*n* (%)	Stat. Sign. *n* (%)	*n* (%)	Stat. Sign. *n* (%)		*n* (%)	Stat. Sign. *n* (%)	n (%)	Stat. Sign. *n* (%)	
T0 VS T1	CD remission achievement by GFD	12 (80%)	10 (66.7%)	0 (0%)	0 (0%)	3 (20%)	7 (77.8%)	4 (44.4%)	0 (0%)	0 (0%)	2 (22.2%)	19 (79.2%)	14 (58.3%)	0 (0%)	0 (0%)	5 (20.8%)
T1 VS T2	prolonged GFD (Ni load?)	0 (0%)	0 (0%)	12 (80%)	5 (33.3%)	3 (20%)	0 (0%)	0 (0%)	8 (88.9%)	5 (55.5%)	1 (11.1%)	0 (0%)	0 (0%)	20 (83.3%)	10 (41.7%)	4 (16.7%)
T2 VS T3	effects of LNiD during prolonged GFD	12 (80%)	7 (46.7%)	2 (13.3%)	0 (0%)	1 (6.7%)	8 (88.9%)	3 (33.3%)	0 (0%)	0 (0%)	1 (11.1%)	20 (83.4%)	10 (41.7%)	2 (8.3%)	0 (0%)	2 (8.3%)
T0 VS T2	active CD VS prolonged GFD (Ni load?)	7 (46.7%)	0 (0%)	5 (33.3%)	0 (0%)	3 (20%)	2 (22.2%)	0 (0%)	5 (55.6%)	0 (0%)	2 (22.2%)	9 (37.5%)	0 (0%)	10 (41.7%)	0 (0%)	5 (20.8%)
T0 VS T3	Active CD VS prolonged GFD + LNiD	12 (80%)	7 (46.7%)	2 (13.3%)	0 (0%)	1 (6.7%)	5 (55.6%)	1 (11.1%)	2 (22.2%)	0 (0%)	2 (22.2%)	17 (70.8%)	8 (33.3%)	4 (16.7%)	0 (0%)	3 (12.5%)
T1 VS T3	initial GFD VS prolonged GFD + LNiD	4 (26.7%)	0 (0%)	6 (40%)	0 (0%)	5 (33.3%)	1 (11.1%)	0 (0%)	7 (77.8%)	0 (0%)	1 (11.1%)	5 (20.8%)	0 (0%)	13 (54.2%)	0 (0%)	6 (25%)

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
