# Peer review of "Beneficial Effects of a Low-Nickel Diet on Relapsing IBS-Like and Extraintestinal Symptoms of Celiac Patients during a Proper Gluten-Free Diet: Nickel Allergic Contact Mucositis in Suspected Non-Responsive Celiac Disease"

_nutrients, 2020, doi:10.3390/nu12082277_

Round 1

Reviewer 1 Report

None

Author Response

Thanks for appreciating our work. 

Reviewer 2 Report

This observational research is an attempt to address a challenging and rather frequent clinical issue- celiac patients with ongoing GI symptoms, despite good adherence to gluten free diet as evident by negative serology and normal histology.  The authors present a novel yet intriguing approach, hypothesizing that intake of nickel-rich diet, causing an allergic contact mucositis (ACM), is a key factor in these patients.  The authors aims included testing patients with refractory celiac disease for nickel ACM, and to describe the effect of a low nickel diet on these patients.

Specific comments:

Introduction:

1.  Lines 57-59.  High-positive serology in patients with typical symptoms may be suffice for formal celiac diagnosis.

2.  Nearly all citations of the scientific background regarding the concept of nickel-ACM and correlations between high-content nickel diet and clinical symptoms are self-citations.  Referencing other authors will strengthen the study rationale.

3.  Introduction is too wordy.

Material & Methods:

The clinical significance of high nickel diet is controversial, especially, given that measuring nickel content in the diet is practically impossible.  Good methodology is essential for a proof of concept. 

4.  Patients with refractory celiac disease were tested for nickel-ACM.  The study would benefit greatly by comparing the study population nickel-ACM prevalence to that in non-refractory celiac patients and in healthy subjects.

5.  High number of patients were excluded from the study due to lactose intolerance (figure 2b and text).  How was this determined?

6. Line 144.   What were the standard instructions given to patients for gluten free and low-nickel diet?   It seems to be an exceptionally challenging diet.

Results:

7.  All remaining patients in the study tested positive for nickel-ACM by omPT.  This is indeed surprising, supporting the request to test non-refractory celiac patients and healthy subjects (see section 4).

8.  Line 181:  ....additional gastrointestinal and/or extra....symptoms have been recorded.  Either specify the results or omit this sentence. 

9. Figures are miss-numbered and one is missing.

10. Line 187.  No clear distinction was made in regard to patients vs. symptoms up to this point.  Should probably state this clearly in the begning of the Results section.

11. Line 189.  What were the typical symptoms to relapse?

12. Figures and text- i.e. lines 187 + 189.  If results were not statistically significant- cannot safely state that improvement was noted; at most- a trend was demonstrated.

Discussion:

13. Discussion is too wordy. 

14. Please comment on the potential placebo effect of close contact with patients during follow-up, including extensive dietary discussions.  

Author Response

Introduction:

  1. Lines 57-59. High-positive serology in patients with typical symptoms may be suffice for formal celiac diagnosis.

Appropriate specifications have been made: “CD diagnosis in adults is usually based on…”.

We have preferred to report what is written in the most recent guidelines, although we fully agree with you: in fact, in pediatric cases, according to the ESPGHAN guidelines, a diagnosis of celiac disease is already accepted on the basis of a remarkable antibody positivity (EMA positive results, anti-tTG x10 ULN) together with clear related-gluten signs and symptoms and positivity for DQ2 and DQ8, in the absence of histological examination. In a recent work of ours regarding celiac adults we have shown similar data (EMA positive results, anti-tTG x3.5 ULN, gluten-related symptoms). Nevertheless, referees generally consider histological examination indispensable for a correct diagnosis.

  1. Nearly all citations of the scientific background regarding the concept of nickel-ACM and correlations between high-content nickel diet and clinical symptoms are self-citations. Referencing other authors will strengthen the study rationale.

Thank you for your suggestion: we added three more references from other groups which deal with nickel allergic contact mucositis and the effects of LNiD on symptoms.

  1. Introduction is too wordy.

We are sorry. We have simplified and shortened some parts of the introduction section.

Material & Methods:

The clinical significance of high nickel diet is controversial, especially, given that measuring nickel content in the diet is practically impossible. Good methodology is essential for a proof of concept.

Your observation is correct and direct monitoring of the nickel content in ingested foods would be optimal. Unfortunately, as mentioned in the discussion, there is currently no standardized method for this measurement. The only method recognized today is the use of tables that report the estimated average nickel content in certain foods, with the accepted limit of a minimum variability.

  1. Patients with refractory celiac disease were tested for nickel-ACM. The study would benefit greatly by comparing the study population nickel-ACM prevalence to that in non-refractory celiac patients and in healthy subjects.

CD patients with relapsing symptoms were tested for Ni omPT in order to know the prevalence of Ni ACM in that specific category of patients. The interesting comparison in Ni ACM prevalence (tested with Ni omPT) with both celiac patients in actual clinical remission and healthy subjects that you suggest could be the subject of a new dedicated study (larger population, different inclusion criteria, etc). On the other hand, the main objectives of our study were to properly support the LNiD with a specific test and properly reduce the nickel-related symptoms (which is impossible to do with asymptomatic celiac subjects or even healthy subjects).

  1. High number of patients were excluded from the study due to lactose intolerance (figure 2b and text). How was this determined?

Lactose intolerance has been evaluated by means of the Lactose Breath Test and genetic evaluation of lactase-gene polymorphism. Proper references have also been added.

  1. Line 144.   What were the standard instructions given to patients for gluten free and low-nickel diet?   It seems to be an exceptionally challenging diet.

In addition to the conventional indications for a correct GFD, patients are advised to exclude foods with a higher nickel content as far as possible. When this is not possible, patients are advised not to take more than one “risky” high-Nickel food per meal and in small quantities. A wide food rotation with allowed foods is also suggested (e.g. including rice, different kinds of meat, other less risky vegetables, etc..): as described in the text, a good support from specialists and nutritionists is essential for an adequate (and personalized) food plan in this peculiar condition. Our positive results and the good compliance by patients seem to confirm the effectiveness of our strategy.

Results:

  1. All remaining patients in the study tested positive for nickel-ACM by omPT. This is indeed surprising, supporting the request to test non-refractory celiac patients and healthy subjects (see section 4).

In our study such a high prevalence of positive omPT can probably be justified by the specific nickel overload that celiac patients make during proper GFD. As mentioned above, given our encouraging preliminary results, a new study on the prevalence of nickel ACM also in asymptomatic celiac patients in clinical remission, symptomatic non-celiac subjects and even healthy controls may be carried out.

  1. Line 181: ....additional gastrointestinal and/or extra....symptoms have been recorded. Either specify the results or omit this sentence.

Proper corrections have been performed, as suggested: we specified that all patients showed at least one additional gastrointestinal or extra-intestinal systemic symptom (among the ones described in materials and methods section) within 48 hours after Ni omPT. Since the late local/systemic response to nickel can be alternative or additional to the immediate response, these clinical data deserve to be at least mentioned in the general description.

  1. Figures are miss-numbered and one is missing.

Sorry for the inconvenience but unfortunately it was a technical page layout problem due to Word. We have arranged the figures in the correct order again and included the figure that had been wrongly removed by the system.

  1. Line 187. No clear distinction was made in regard to patients vs. symptoms up to this point. Should probably state this clearly in the beginning of the Results section.

The exposure of the results follows a pre-established rational order, as specified in materials and methods section: 1) to exclude non-recruitable subjects; 2) to investigate the prevalence of positive Ni omPT in the remaining population; 3) to study the trend of symptoms in the subjects recruited (patients vs symptoms).

  1. Line 189. What were the typical symptoms to relapse?

Thank you for your question. As reported in Figure 4 and 5, the symptoms which significantly relapsed were abdominal pain, bloating, nausea, swelling, loose stools, dermatitis, fatigue, numbness of the limbs, muscle and joint pain. We also added this specification in the text after your request, to make it clearer for the reader.

  1. Figures and text- i.e. lines 187 + 189. If results were not statistically significant- cannot safely state that improvement was noted; at most- a trend was demonstrated.

As requested, a further distinction between improvement trend and statistically significant improvement was made. Explicit changes have been made in the results section.

Discussion:

  1. Discussion is too wordy.

We shortened discussion section, as requested.

  1. Please comment on the potential placebo effect of close contact with patients during follow-up, including extensive dietary discussions.

Thank you for your suggestion. We added a dedicated paragraph at the end of discussion section.

Reviewer 3 Report

The article of Borghini et al. is an original research enquiring about the prevalence and the role of Nickel-related allergic contact mucositis in non-responder celiac patients. The idea is novel and original with potential practical implications but some major flaws should be revised.

  1. Title: The title is misleading. In fact, patients are clinically non responders to GFD but there is no reference at persistence of malabsorption or atrophy in those patients. Also because, in that setting, in the suspect of refractory celiac disease, a completely different diagnostic pathway should be performed including molecular & flow cytometry analysis, capsule endoscopy, radiologic exams and eventually device-assisted enteroscopy in order to exclude a real refractoriness to diet. Indeed, in the paper the comparison is with other concomitant gastrointestinal conditions as HP infections, lactose intolerance, IBS-like symptoms. Thus, on the one hand, the title should be changed and adhere more strictly to the main conclusions of the paper. On the other hand, a thorough explanation of how RCD is excluded should be provided in the manuscript. 
  2. Abstract: In the conclusions, the role of alimentary Ni intake should not be compared to gluten contaminations (line 42) that should be always be excluded in the persistence of gastrointestinal symptoms, being a cause of persistent atrophy and higher risk of complicated celiac disease.  Conversely, the role of Ni should be take into account in case of non responders celiac patients only after the exclusion of RCD and gluten contaminations. 
  3. Introduction: the main references are self-referential (see Ref 1,2,3,7,8,9,10). I suggest to identify some more relevant study in literature (es. Green et al, Murray et al, Elli et al) on celiac disease and its gastrointestinal/extraintestinal symptoms. 
  4. Introduction: the data of “30%” of estimated prevalence is not so clearly referenced (maybe “according to the European Surveillance System of Contact Allergy (ESSCA), the prevalence of epicutaneous patch test positive to Ni is about 30% in Europe” with the updated reference?)
  5. Methods: The employ of omPT has been externally validated in other studies/experiences? If so, it should be cited.
  6. Methods: The study design is defined “observational” but it is questionable as a new dietary regimen is undertaken after an evaluation with trained dietician in the 20 enrolled patients (Pilot study?). 
  7. Results: line 170 “27 patients were excluded”. It should be 17 as in the Figure 2. 
  8. Figure 3-4-5: there was probably an uploading error because it is the same of Figure 2 and the other ones did not correspond to the legend
  9. Discussion: All the 20 patients studied showed a Ni omPT positive results. A 100% prevalence of the disorders, even if partially explained in the Discussion, is one of the major bias of the study. In particular, a “real-life” prevalence of 23.5% is defined (line 40 of discussion): the 85 symptomatic CD patients were all tested and negative? Otherwise, the percentage should not be expressed. Analogously, if negative, they could be used as a “control” group for the dietary approach.  
  10. Discussion: a comment about the degree of acceptance and compliance of patients when facing two different dietary regimen would be interesting

Author Response

The article of Borghini et al. is an original research enquiring about the prevalence and the role of Nickel-related allergic contact mucositis in non-responder celiac patients. The idea is novel and original with potential practical implications but some major flaws should be revised.

Title: The title is misleading. In fact, patients are clinically non responders to GFD but there is no reference at persistence of malabsorption or atrophy in those patients. Also because, in that setting, in the suspect of refractory celiac disease, a completely different diagnostic pathway should be performed including molecular & flow cytometry analysis, capsule endoscopy, radiologic exams and eventually device-assisted enteroscopy in order to exclude a real refractoriness to diet. Indeed, in the paper the comparison is with other concomitant gastrointestinal conditions as HP infections, lactose intolerance, IBS-like symptoms. Thus, on the one hand, the title should be changed and adhere more strictly to the main conclusions of the paper. On the other hand, a thorough explanation of how RCD is excluded should be provided in the manuscript.

Thank you for your comment. As suggested, we changed the title “…Nickel Allergic Contact Mucositis in Suspected Non-Responsive Celiac Disease Rather Than Refractory Celiac Disease?”. Indeed, the definition of non-responsive celiac disease is more correct than refractory celiac disease, also because all the patients in the study no longer have evidence of intestinal atrophy. We also included reference 5 in the text after your suggestion [5. Penny HA, Baggus EMR, Rej A, Snowden JA, Sanders DS. Non-Responsive Coeliac Disease: A Comprehensive Review from the NHS England National Centre for Refractory Coeliac Disease. Nutrients. 2020 Jan 14;12(1):216.]

Abstract: In the conclusions, the role of alimentary Ni intake should not be compared to gluten contaminations (line 42) that should be always be excluded in the persistence of gastrointestinal symptoms, being a cause of persistent atrophy and higher risk of complicated celiac disease. Conversely, the role of Ni should be take into account in case of non responders celiac patients only after the exclusion of RCD and gluten contaminations.

As suggested, we properly changed this part as follows: “the recrudescence of gastrointestinal and extra-intestinal symptoms observed in CD subjects during GFD may be due to the increase in alimentary Ni intake, once gluten contamination and persisting villous atrophy are excluded”.

Introduction: the main references are self-referential (see Ref 1,2,3,7,8,9,10). I suggest to identify some more relevant study in literature (es. Green et al, Murray et al, Elli et al) on celiac disease and its gastrointestinal/extraintestinal symptoms.

As suggested, additional relevant references have been added.

Introduction: the data of “30%” of estimated prevalence is not so clearly referenced (maybe “according to the European Surveillance System of Contact Allergy (ESSCA), the prevalence of epicutaneous patch test positive to Ni is about 30% in Europe” with the updated reference?).

Thank you for your comment. We performed the correction, as suggested.

Methods: The employ of omPT has been externally validated in other studies/experiences? If so, it should be cited.

The use of Ni omPT has so far been studied in multiple studies of our research group only, over the past ten years, deepening not only the clinical-symptomatological aspect: immunohistochemistry tests were performed on oral biopsies, as well as peripheral blood lymphocyte typing and laser doppler perfusion imaging to support its usefulness and reliability. All the most important bibliographic sources in this regard have been included.

Methods: The study design is defined “observational” but it is questionable as a new dietary regimen is undertaken after an evaluation with trained dietician in the 20 enrolled patients (Pilot study?).

Thank you for your comment, we agree with you. We switched “observational” with “pilot” study.

Results: line 170 “27 patients were excluded”. It should be 17 as in the Figure 2.

Thank you for your correction, it was a typing error.

Figure 3-4-5: there was probably an uploading error because it is the same of Figure 2 and the other ones did not correspond to the legend

Sorry for the inconvenience but unfortunately it was a technical page layout problem due to Word. We have arranged the figures in the correct order again and included the figure that had been wrongly removed by the system.

Discussion: All the 20 patients studied showed a Ni omPT positive results. A 100% prevalence of the disorders, even if partially explained in the Discussion, is one of the major bias of the study. In particular, a “real-life” prevalence of 23.5% is defined (line 40 of discussion): the 85 symptomatic CD patients were all tested and negative? Otherwise, the percentage should not be expressed. Analogously, if negative, they could be used as a “control” group for the dietary approach.

Only those 20 out of 85 symptomatic patients were tested for Ni omPT as their symptomatology could not be justified by other disorders (e.g. lactose intolerance, HP infection,…) after serological/histological remission of CD had been ascertained. This, as specified in the text, is a limitation of being a pilot study and it would explain such a high percentage (100%) of positive omPTs.

This is the reason why, in order not to offer misleading data, we stated in discussion section that those 20 patients with Ni omPT positive results should be contextualized among the 85 still symptomatic CD patients, reporting a more realistic prevalence of at least 23.5%.

On the other hand, considering the possible overlap among IBS-like disorders within the same patient, the exclusion of those 62 patients guarantees integrity of the study (strict exclusion criteria applied) and absence of confounding factors in the following clinical evaluation.

As suggested, based on this experience, future clinical trials may be conducted to more properly study Ni ACM prevalence in all subjects initially recruited, as well as in controls.

Discussion: a comment about the degree of acceptance and compliance of patients when facing two different dietary regimen would be interesting

At the end of the discussion section we included a paragraph about the risks and benefits of close clinical support during our study, also as suggested by reviewer 2. After your interesting suggestion, we included how close contact has most likely facilitated the management of two strict dietary regimens such as GFD and LNiD, in light of our encouraging clinical results and the very low number of drop-outs, although the study has been very long.

Round 2

Reviewer 3 Report

The article of Borghini et al. is an original research enquiring about the prevalence and the role of Nickel-related allergic contact mucositis in non-responder celiac patients. After the reviewers' suggestion, a thorough revision has been made and, in my view, no more corrections are needed.